# A Real-World Retrospective Analysis of the Management of Advanced Urothelial Carcinoma in Canada

**Feras A. Moria [1], Changsu L. Park [1], Bernhard J. Eigl [2], Robyn Macfarlane [3], Michel Pavic [4]** and **Ramy R. Saleh [1,***

[1] McGill University Health Center, Montreal, QC H4A 3J1, Canada; feras.moria@mail.mcgill.ca (F.A.M.); changsu.park@mail.mcgill.ca (C.L.P.)
[2] BC Cancer Vancouver Centre, Vancouver, BC V5Z 4E6, Canada; bernie.eigl@bccancer.bc.ca
[3] QEII Health Sciences Centre, Halifax, NS B3H 2Y9, Canada; robynj.macfarlane@nshealth.ca
[4] Centre Hospitalier Universitaire de Sherbrooke, Sherbrooke, QC J1H 5N4, Canada; michel.pavic@usherbrooke.ca
[*] Correspondence: ramy.saleh@mcgill.ca; Tel.: +1-(514)-934-1934 (ext. 35800)

**Abstract:** Locally advanced or metastatic urothelial carcinoma (aUC) presents a significant challenge with high mortality rates. Platinum-based chemotherapy remains the established frontline standard of care, and a switch-maintenance strategy with immunotherapy has now emerged as a new standard for aUC patients without disease progression, following initial platinum therapy. Examining the treatment patterns is imperative, given the evolving therapeutic landscape. In this study, we conducted a retrospective medical chart review of 17 Canadian oncologists treating patients with aUC to assess unmet needs in Canadian aUC patient care. Data from 146 patient charts were analyzed, revealing important clinical insights about the management of aUC. A substantial proportion of patients (53%) presented with de novo metastatic disease, which was possibly influenced by pandemic-related care disruptions. Variability was evident in the cisplatin eligibility criteria, with a majority (70%) of oncologists utilizing a 50 mL/min threshold. Most favored four cycles of platinum-based chemotherapy to spare the bone marrow for future therapies and prevent patient fatigue. Notably, some eligible patients were kept under surveillance rather than receiving maintenance therapy, suggesting a potential gap in awareness regarding evidence-based recommendations. Furthermore, managing treatment-related adverse events was found to be one of the biggest challenges in relation to maintenance immunotherapy. In conclusion, our findings provide the first comprehensive overview of aUC treatment patterns in Canada following the approval of maintenance immunotherapy, offering insights into the decision-making process and underscoring the importance of evidence-based guidelines in aUC patient management.

**Keywords:** urothelial carcinoma; maintenance therapy; avelumab; immunotherapy; platinum-based chemotherapy; cisplatin; carboplatin; chart review; care gap; unmet need

## 1. Introduction

Bladder cancer is the sixth most common cancer worldwide [1]. In Canada, the estimated incidence for 2022 was 13,300 new cases, with 2500 disease-related deaths per year [2]. The disease is more common in the elderly, with an average age of diagnosis of between 70 and 84 years [3], and disproportionally affects males (3:1 male-to-female ratio) [4]. The most important risk factor for developing bladder cancer is tobacco smoking, accounting for ~50% of cases [5], followed by exposure to occupational carcinogens [6]. Current smokers have a four- to five-fold increased relative risk of developing bladder cancer than non-smokers [7].

Bladder cancers are often referred to as urothelial carcinomas (UC) as most tumors arise from the urothelial cells lining the bladder and urinary tract [8]. They are further classified as non-muscle-invasive bladder cancer (NMIBC) or muscle-invasive bladder cancer (MIBC), with muscle-invasive tumors having a more aggressive disease biology. Between 10 and

25% of patients with NMIBC will progress to MIBC or metastatic disease and between 40 and 50% of those with early-stage MIBC will relapse after initial treatment [9,10]. Moreover, approximately 15% of patients have locally advanced or metastatic urothelial carcinoma (aUC) at presentation [10]. MIBC and aUC are also associated with higher mortality rates. The five-year survival rates stand at 36% to 48% for localized MIBC, contrasting with a mere 5% for those with aUC [4], underscoring a significant unmet demand for efficacious therapeutic interventions in the metastatic setting.

In recent years, immunotherapies have revolutionized the treatment of solid tumors, including aUC, offering durable responses with favorable safety profiles. The immune checkpoint inhibitor pembrolizumab has been approved for second-line use, based on the positive data from KEYNOTE-045 [11]. Additionally, immunotherapies have been approved for specific patients who are ineligible for platinum-based treatments [12,13]. Novel immunotherapy combinations are emerging in the frontline setting, with the US Food and Drug Administration recently granting accelerated approval for pembrolizumab plus enfortumab-vedotin for those who are ineligible for cisplatin-based chemotherapy, based on data from a randomized phase II trial [14]. Although immunotherapies have emerged as a significant advancement, platinum-based chemotherapy regimens continue to serve as the established frontline standard of care, primarily due to the inherent chemosensitivity of the disease. Among these, the preferred regimen is a combination of gemcitabine and cisplatin [10,12]. The objective response rates (ORR) are 40 to 50% with frontline chemotherapy. However, the durability of the response is modest, with median progression-free survival (PFS) of approximately 8 months and median overall survival (OS) of only 13 to 15 months with cisplatin-based regimens (and 9 to 10 months with carboplatin-based regimens) [15–18]. Furthermore, the cumulative toxicities of chemotherapy limit the potential duration of treatment. In this context, a switch-maintenance strategy with immunotherapy constitutes an attractive approach to maximize progression-free survival.

Maintenance immunotherapy with avelumab was investigated in a randomized, open-label phase III trial (JAVELIN Bladder 100), evaluating the efficacy and safety of avelumab plus the best supportive care (BSC) in adult patients with histologically confirmed, unresectable, locally advanced or metastatic urothelial carcinoma. First reported in 2020 [19] and updated in 2023, with ≥2 years of follow-up in all study patients [20], avelumab significantly improved OS by close to 9 months compared to BSC (median OS of 23.8 months [95% CI, 19.9–28.8] with avelumab + BSC, vs. 15.0 months [95% CI, 13.5–18.2] with BSC alone) with a hazard ratio of 0.76 (95% CI, 0.631–0.915). It is the first and only drug to improve OS as maintenance therapy after frontline platinum-based chemotherapy in patients with aUC and the first drug to be approved in this setting. It has since been incorporated as a standard of care for the first-line maintenance treatment of patients with aUC whose disease has not progressed after first-line platinum-based chemotherapy [12,13].

Considering the expanding therapeutic options and new treatment paradigms in aUC, it is important to examine treatment patterns among patients with aUC over this initial period of approval to assess how first-line maintenance therapy is being used in routine clinical practice and its impact on upstream and downstream treatments. There are currently limited data related to treatment patterns in aUC and the adoption of first-line maintenance therapy in clinical practice outside of the United States, with no data for Canada. In this study, we sought to characterize real-world treatment patterns by conducting a retrospective medical chart review of oncologists across Canada who treat aUC, to assess unmet needs in Canadian patients with aUC.

## 2. Materials and Methods

### 2.1. Study Design and Analysis

This study is based on a retrospective medical chart review of Canadian medical oncologists with a clinical practice in bladder cancer. To collect de-identified patient data for the chart review, an online questionnaire was developed by the authors of the study, all of whom are Canadian medical oncologists renowned for their expertise in bladder

cancer. The questionnaire was designed to capture (1) demographic information about participants, (2) de-identified patient-level data, including the relevant medical history, and (3) retrospective data about the pharmacological management of each patient's aUC. Each participant was invited to conduct a chart review on ten patients in their clinical practice. The full chart review questionnaire can be found in Supplementary Table S1.

The invitation period was open from December 2022 to January 2023. During this period, the participants were briefed by the authors of the study about the study rationale and objectives and received a demonstration of how to complete the online chart review. Data were collected from 26 January 2023 to 22 March 2023. This collection period was selected to ensure that clinical management was up to date as it pertained to the post-approval adoption of maintenance therapy.

The study's primary objectives were to outline baseline demographic and clinical attributes, as well as elucidate the patterns of first-line treatment. Additionally, its secondary objectives involved delving into treatment priorities, gaining insight into the practical application of cisplatin eligibility criteria, examining factors that impact the administration of cycles of platinum-based chemotherapy, and identifying the obstacles encountered by clinicians when managing patients undergoing maintenance therapy.

Descriptive statistics were presented as counts and percentages for categorical variables and as means, with standard deviation for continuous variables.

### 2.2. Patient Criteria

To be eligible for the chart review, patients had to meet the following four criteria:

1. Patients must have been diagnosed with unresectable locally advanced or metastatic urothelial carcinoma (aUC), either de novo aUC or as a progression from an earlier stage of UC.
2. The diagnosis must have been no earlier than January 2021, to coincide with the availability of new therapeutic options.
3. Patients must have received or completed, at the time of data entry, at least one systemic therapy, which should have included first-line platinum-based chemotherapy.
4. Patients should not have shown disease progression during first-line platinum-based chemotherapy or within 12 months of neoadjuvant or adjuvant treatment.

### 2.3. Focus Groups

To facilitate interpretation of the aggregated findings, qualitative data were acquired by conducting focus groups involving the study participants. These focus groups, each spanning 90 min, were conducted over the period of 12–19 April 2023. The investigators of the study conducted the focus groups, with carefully designed discussion questions intended to extract meaningful perspectives. The objectives of the focus groups were to review and solicit insights on the aggregated results from the chart review and to capture additional real-world perspectives about the management of patients with aUC in Canada within the context of the latest evidence and guideline recommendations. The focus groups were recorded and the proceedings were professionally transcribed. Subsequently, the research team compiled a report that distilled the key insights, which was then shared with the participants. For reference, a copy of the discussion guide is found in Supplementary Table S2.

### 3. Results

#### 3.1. Participants

3.1.1. Oncologist Demographics

A total of 17 oncologists participated in the study (see Table 1). Most oncologists (94%) indicated that they were medical oncologists, with one participant being a hematologist and oncologist. The sample included geographic representation across 7 Canadian provinces, with Ontario, Quebec, and British Columbia being the most well-represented regions. Approximately half of the sample (47%) had been practicing in oncology for more than

10 years. University and academic medical centers (82%) represented the most common principal practice types. Most oncologists (88%) had an average of 3 to 10 new aUC consultations each month.

**Table 1.** Oncologist demographics.

| Characteristics (*n* = 18) | *n* | % |
|---|---|---|
| **Oncology specialty** | | |
| Medical oncologist | 16 | 94 |
| Hematologist oncologist | 1 | 6 |
| **Province of practice** | | |
| Ontario | 5 | 29 |
| Quebec | 4 | 24 |
| British Columbia | 4 | 24 |
| Alberta | 1 | 6 |
| New Brunswick | 1 | 6 |
| Nova Scotia | 1 | 6 |
| Saskatchewan | 1 | 6 |
| **Primary setting of practice** | | |
| Academic-/university-affiliated hospital | 14 | 82 |
| Community-based hospital/clinic/practice | 3 | 18 |
| **Years practicing oncology** | | |
| 0–10 | 9 | 53 |
| 11–20 | 5 | 29 |
| 21–30 | 2 | 12 |
| More than 30 | 1 | 6 |
| **Number of new aUC consultations in the last month** | | |
| 0 | 0 | 0 |
| 1–2 | 2 | 12 |
| 3–5 | 8 | 47 |
| 5–10 | 7 | 41 |
| More than 10 | 0 | 0 |
| **Creatinine clearance threshold for cisplatin eligibility** | | |
| 60 mL/min | 3 | 18 |
| 55 mL/min | 1 | 6 |
| 50 mL/min | 12 | 70 |
| 45 mL/min | 1 | 6 |

### 3.1.2. Creatinine Clearance Threshold for Cisplatin Eligibility

Insights about the creatinine clearance (CC) threshold when assessing eligibility for first-line cisplatin treatment were collected. The majority (70%) of participants use a CC threshold of 50 mL/min. A minority of participants (6%) use an even lower threshold. Only a small percentage (18%) adhere to the Galsky criteria CC threshold of 60 mL/min, which has been widely used in clinical trials to determine cisplatin eligibility [21]. When probed about this issue in the focus groups, the participating oncologists said that in clinical practice, the Galsky criteria are simply used to guide treatment selection, but that exceptions should be considered. They reasoned that high-volume treaters have a greater tolerance for lower CC thresholds. Finally, the most common (75%) method to assess kidney function was an estimated CC using the Cockroft–Gault variant (Table S3 in the Supplementary Materials). Kidney function was only measured instead of estimated in a minority of cases (5%).

*3.2. Patient Baseline Demographics and Clinical Charactersistics*

3.2.1. Demographics

A total of 146 patient chart reviews were completed (Table 2). Most patients were male (73%) and the median age at the time of diagnosis for aUC was 71 years, which is consistent with known aUC demographics [3]. Most patients were diagnosed with aUC in 2021 or 2022 (91%). Two-thirds of patients (64%) had baseline comorbidities, with the most common being hypertension (47%), dyslipidemia (32%), chronic kidney disease (21%), diabetes (20%), and atherosclerotic cardiovascular disease (14%). Most patients (74%) had a history of smoking, with many (21%) still currently smoking.

**Table 2.** Selected patient baseline demographics and clinical characteristics.

| Characteristic (*n* = 146) | *n* | % |
|---|---|---|
| Male | 106 | 73 |
| Female | 40 | 27 |
| Age, years; median (range) | 71 (38–90) | |
| **Year of diagnosis of aUC** | | |
| 2021 | 59 | 40 |
| 2022 | 75 | 51 |
| 2023 | 12 | 8 |
| **Comorbidities** | 93 | 64 |
| Hypertension | 68 | 47 |
| Dyslipidemia | 46 | 32 |
| Chronic kidney disease | 30 | 21 |
| Diabetes | 29 | 20 |
| Atherosclerotic cardiovascular disease | 20 | 14 |
| Obesity/overweight | 8 | 5 |
| Gastrointestinal disease | 6 | 4 |
| Heart failure | 4 | 3 |
| Hearing loss | 4 | 3 |
| Other [1] | 50 | 34 |
| No comorbidities | 53 | 36 |
| **History of smoking** | | |
| Yes, current | 31 | 21 |
| Yes, former | 77 | 53 |
| No | 38 | 26 |
| **Medical history** | | |
| De novo aUC | 77 | 53 |
| Progression from an earlier stage of UC | 69 | 47 |
| **Symptoms leading de novo aUC diagnosis (*n* = 77)** | | |
| Painless gross hematuria | 19 | 25 |
| Irritative bladder symptoms [2] | 9 | 12 |
| Other [3] | 11 | 14 |
| **Cystectomy if progression from earlier stage (*n* = 69)** | | |
| Yes | 32 | 46 |
| No | 37 | 53 |

**Table 2.** *Cont.*

| Characteristic (*n* = 146) | *n* | % |
|:---:|:---:|:---:|
| **Metastatic sites** | | |
| Lymph node beyond the common iliacs | 99 | 68 |
| Lung | 55 | 38 |
| Bone | 32 | 22 |
| Liver | 20 | 14 |
| Other [4] | 14 | 10 |
| None (locally advanced unresectable disease) | 3 | 2 |
| *FGFR3* **mutation** | | |
| Not tested [5] or unknown [6] | 87 | 60 |
| Positive | 6 | 4 |
| Negative | 53 | 36 |

[1] Other comorbidities include atrial fibrillation (*n* = 3) psychiatric disorder (*n* = 3), gout (*n* = 3), unrelated malignancies (early-stage lung cancer, Merkel cell carcinoma, and a thyroid nodule) (*n* = 3), COPD (*n* = 2), rheumatoid arthritis (*n* = 2), alcohol-related neuropathy, asymptomatic bradycardia, asthma, cognitive impairment due to head injury, cholecystectomy, eczema, gastric bypass, heart block with a pacemaker, interstitial lung disease, mild cognitive impairment, NASH, obstructive sleep apnea, possible seizure disorder, paralysis secondary to an MVA, osteoarthritis, osteoporosis, peripheral vascular disease, primary sclerosing cholangitis, psoriasis, stroke, and umbilical hernia repair. [2] Irritative bladder symptoms include dysuria, urgency, and frequency of urination. [3] Other symptoms leading to the suspicion of aUC include pelvic or bone pain (*n* = 3), flank pain (*n* = 2), palpable mass on examination (*n* = 2), incidental finding on imaging (*n* = 2), lower extremity edema (*n* =1) and constitutional symptoms (*n* = 1). [4] Other metastases include the vulva, renal bed, omental metastasis, retroperitoneal lymph nodes, para-aortic lymph nodes, ascitic fluid, retroperitoneal lymph nodes, adrenal gland, peritoneal carcinomatosis, and psoas. [5] Checking for *FGFR3* mutation was not part of routine testing for 71 patients (49%). [6] Unknown signifies that the results were pending at the time of data entry. Abbreviations: COPD = chronic obstructive pulmonary disease; MVA = motor vehicle accident; NASH = non-alcoholic steatohepatitis.

### 3.2.2. De Novo aUC vs. Metastatic Progression from Early-Stage Disease

There was an even distribution of patients with de novo aUC (53%) compared to patients who had experienced a metastatic progression from an earlier stage of UC. Respondents from Western Canada selected more patients with de novo metastatic disease, whereas respondents from Eastern Canada selected more patients who had recurrence from an earlier stage of disease.

### 3.2.3. Diagnostic Workup

We investigated the diagnostic workup of all patients. The most common steps in the initial evaluation of aUC were a history and a physical examination (90%), a biopsy (54%), transurethral resection of a bladder tumor (TURBT) (51%), and cystoscopy with standard white light (50%). The most common imaging modalities were a computed tomography (CT) urogram (39%), chest imaging (88%), and a bone scan (50%). These findings are consistent with recent clinical practice guidelines for diagnosis [12].

In patients with de novo metastatic disease (*n* = 77), the most common symptoms that led to the suspicion of aUC were painless gross hematuria (25%) and irritative bladder symptoms (12%), which is consistent with the most common symptoms at presentation that are reported in the literature [4]. In patients who had progressed from an earlier stage of disease (*n* = 69), the primary tumor (T) stage at the initial diagnosis of localized disease was distributed between stages T1 (25%), T2 (36%), and T3 (33%) (Table S3 in the Supplementary Materials). Most patients (81%) who had progressed from an earlier stage of disease also had no regional lymph node (N0) metastases at the time of the initial diagnosis of early-stage disease (Table S3). When respondents were queried about this finding during the focus groups, many were surprised, as lymph node involvement is a risk factor for disease progression. A possible explanation is a discrepancy between clinical staging and pathological staging. The chart review questionnaire did not make a distinction between the two. In terms of metastatic sites at the time of the aUC workup, lymph nodes beyond the common iliacs were the most common location (68%), followed by the lungs (38%), bones (22%), and liver (14%).

### 3.2.4. Cystectomy

We also looked at the proportion of patients who had undergone radical cystectomy (RC), namely, those in the subgroup who had progressed from early-stage disease (*n* = 69). Less than half (46%) of these patients had undergone an RC for early-stage disease. When queried about this finding during the focus groups, participants expected a higher proportion, given that the standard therapy for localized MIBC, as recommended by the Canadian Urological Association guidelines, is RC (Level 1, strong recommendation) [22]. They further indicated that in practice, cystectomies are typically favored in patients under 75 years of age. Bladder-sparing practices, usually involving trimodality therapy (Level 3, moderate recommendation), are offered to selected patients wishing to preserve their bladder, those unfit for a cystectomy, or those refusing a cystectomy [22]. A likely explanation for the low percentage of patients who underwent an RC is the selection bias for patients who failed local treatment. Ultimately, this limits what we can infer about early-stage treatment practices.

### 3.2.5. Molecular Testing

Molecular testing was performed in half of the patients (51%). Of these patients, very few (4%) had fibroblast growth factor receptor (*FGFR*) 3 genetic alterations, with the majority having no alterations (36%). The low *FGFR3* positivity rate compared to previous reports—15–20% *FGFR3* positivity for invasive UC [23,24]—may reflect the sampling bias for high-risk patients, who are purported to have a lower incidence of *FGFR3* alterations. *FGFR3* alterations are generally associated with a lower grade and stage among all urothelial bladder carcinomas [25]. For example, among the T1 tumors, *FGFR3* expression was shown to be associated with lower-grade tumors and a lower risk of disease progression [26]. Even so, how *FGFR3* alterations correlate with outcomes in an advanced setting remains unclear. Nevertheless, many participating oncologists shared the opinion that a positivity rate of below 10% reflected their clinical experience.

Molecular testing was notably absent in half (49%) of the patient cohort, and none of the patients underwent PD-L1 expression testing. Upon probing for the factors contributing to the absence of testing, participants pointed to limited access to testing as the primary issue. Additionally, it was observed that in certain cases, due to the retrospective nature of patient entries, testing was not conducted due to factors such as the unavailability of *FGFR3* testing, patient death before testing, or insufficient tissue samples for testing. Adjusting for these variables, it is plausible that the frequency of molecular testing has increased since then.

The focus group participants underscored the variability in testing accessibility throughout different regions of Canada. Notably, reflex testing is not universally adopted across most provinces as a standard practice. Ontario has introduced reflex *FGFR3* testing, although this integration lacks complete systematization due to persistent hurdles in the testing process. Challenges such as tissue scarcity and prolonged delays continue to impede the testing procedure.

### 3.3. Patient Management

### 3.3.1. Subgroups

Patients were distributed into four broad subgroups, based on their status at the index date (Table 3): (1) currently on first-line platinum-based chemotherapy (25%); (2) currently under surveillance following first-line platinum-based chemotherapy (10%); (3) currently on maintenance therapy or having recently completed first-line platinum-based chemotherapy and starting maintenance soon (46%); and (4) progressive disease (i.e., receiving second-line therapy or beyond) (19%). Table 3 summarizes the data by these subgroups since the chart review questionnaire had specific questions for each of these subgroups.

### 3.3.2. Treatment Goals

When questioned about their priority treatment goals, most participants sought to improve OS (95%), prolong PFS (62%), improve the quality of life (62%), and reduce the burden of disease (32%) (Table S4 in the Supplementary Materials). When further queried about the reasons for the selection of first-line therapy, the top reasons were that it was the most efficacious option for this patient (62%), it was a tolerable option (32%) with a favorable safety profile (29%), and the patient was eligible for cisplatin (29%) (Table S4).

**Table 3.** Patient management components.

| Component, *n* (%) | All | 1 L ChT (*n* = 37) | Surveillance (*n* = 14) | 1 L MT 1 (*n* = 67) | 2 L + Tx (*n* = 28) |
|---|---|---|---|---|---|
| **1 L ChT (*n* = 146)** | | | | | |
| Gem-cis (GC) | 74 (51) | 15 (40) | 3 (21) | 38 (57) | 18 (64) |
| Gem-carb | 68 (47) | 21 (57) | 10 (72) | 28 (42) | 10 (36) |
| MVAC | 0 | 0 | 0 | 0 | 0 |
| ddMVAC | 2 (1) | 0 | 1 (7) | 1 (1) | 0 |
| Other [2] | 1 (1) | 1 (3) | 0 | 0 | 0 |
| **Response to 1 L ChT (*n* = 109)** | | | | | |
| Disease progression | 9 (8) | - | 5 (37) | 1 (1) | 3 (11) |
| Stable disease | 24 (22) | - | 3 (21) | 12 (18) | 9 (32) |
| Partial response | 65 (60) | - | 3 (21) | 46 (69) | 16 (57) |
| Complete response | 11 (10) | - | 3 (21) | 8 (12) | 0 |
| **Received MT, *n*** | 85 | - | - | 67 | 18 |
| MT-emergent AEs that were difficult or time-consuming to manage | 8 (9) | - | - | 5 (7) [3] | 3 (17) [4] |
| **Timeframe between ChT and MT (*n* = 122)** | | | | | |
| <4 weeks | 15 (12) | 6 (16) [5] | - | 7 (10) | 2 (11) |
| 4–6 weeks | 64 (53) | 23 (62) [5] | - | 32 (48) | 9 (50) |
| 6–8 weeks | 28 (23) | 5 (14) [5] | - | 18 (27) | 5 (28) |
| 8–10 weeks | 15 (12) | 3 (8) [5] | - | 10 (15) | 2 (11) |
| **Duration of MT, median (range) (*n* = 18) [6]** | | - | - | - | 6 (<1–19) |
| **Reason for discontinuing MT (*n* = 18) [6]** | | | | | |
| Disease progression | 17 (94) | - | - | - | 17 (94) |
| Patient preference | 1 (6) | - | - | - | 1 (6) |
| **2 L therapy (*n* = 28)** | | - | - | - | |
| Pembrolizumab or immunotherapy alternative (avelumab, durvalumab) | 11 (39) | - | - | - | 11 (39) |
| Enfortumab-vedotin | 9 (32) | - | - | - | 9 (32) |
| Reinduction with ChT | 1 (4) | - | - | - | 1 (4) |
| Erdafitinib | 1 (4) | - | - | - | 1 (4) |
| Clinical trial | 1 (4) | - | - | - | 1 (4) |
| Other [7] | 5 (17) | - | - | - | 5 (18) |

**Table 3.** *Cont.*

| Component, *n* (%) | All | 1 L ChT (*n* = 37) | Surveillance (*n* = 14) | 1 L MT 1 (*n* = 67) | 2 L + Tx (*n* = 28) |
|---|---|---|---|---|---|
| **3 L therapy (*n* = 12)** | | - | - | - | |
| Enfortumab-vedotin | 4 (33) | - | - | - | 4 (33) |
| Paclitaxel | 2 (17) | - | - | - | 2 (17) |
| Erdafitinib | 1 (8) | - | - | - | 1 (8) |
| Other [8] | 5 (42) | - | - | - | 5 (42) |

[1] The MT subgroup included patients who had completed first-line ChT and who were starting MT in the coming weeks or were already on MT. [2] One participant indicated GC split-dosing. [3] Infusion-related reactions (*n* = 2) and pruritis (*n* = 3) were the only events reported that were difficult/time-consuming to manage. [4] In the 2 L and beyond subgroup, if MT was withheld at any point, the entry was counted as "difficult/time-consuming to manage"; AEs (*n* = 2) and hospitalization for pain management of bone metastases (*n* = 1) were the reasons for withholding MT. [5] This represents the number of weeks that the participating oncologist plans to start MT after the completion of chemotherapy. [6] Data for the duration of MT and reasons for discontinuing therapy were only captured for those within the subgroup of 2 L therapy and beyond who had received MT (*n* = 18/28). [7] Other patients (*n* = 5) received BSC or palliative care in the 2 L setting; in 2 cases, patient preference was specified as the driving factor. [8] Other 3 L options were not specified. Abbreviations: 1 L = first-line therapy; 2 L = second-line therapy; AEs = adverse events; BSC = best supportive care; ChT = chemotherapy; ddMVAC = dose-dense methotrexate, vinblastine, doxorubicin, and cisplatin; Gem-cis = gemcitabine plus cisplatin; Gem-carb = gemcitabine plus carboplatin; MT = maintenance therapy; MVAC = methotrexate, vinblastine, doxorubicin, and cisplatin.

### 3.3.3. First-Line Chemotherapy

All patients were administered first-line platinum-based chemotherapy, being given either gemcitabine-cisplatin (GC) (51%) or gemcitabine-carboplatin (47%)—only 2 patients (1%) were administered first-line dose-dense methotrexate, vinblastine, doxorubicin, and cisplatin (dd-MVAC). In the focus groups, most participants indicated that they favor 4 cycles of chemotherapy. Maintenance therapy was planned for all patients (100%) who are currently on first-line platinum-based chemotherapy.

Response to first-line chemotherapy was also assessed. In patients who were no longer on first-line platinum-based chemotherapy (*n* = 109), most had experienced a partial response to chemotherapy (60%) or had stable disease (22%), with a minority having experienced a complete response (10%). Nine patients (8%) exhibited progressive disease during first-line chemotherapy. Although progressive disease during first-line chemotherapy was an exclusion criterion, these 9 patients were still included in the subsequent analyses, given the descriptive nature of the study.

### 3.3.4. Surveillance

A small subgroup of patients (*n* = 14, 10%) were under surveillance following first-line chemotherapy. Approximately one-third (36%) had experienced progressive disease, but most (64%) had either stable disease, a partial response, or a complete response to chemotherapy. When queried about the reasons for not administering maintenance therapy (Figure 1), patient preference was the most common reason (29%), followed by pre-existing medical conditions (21%), minimal residual disease following chemotherapy (14%), concerns about the patient's ability to tolerate therapy (14%), poor performance status (14%), being elderly (14%), and the presence of concomitant immunosuppressive therapy (7%). Other reasons were cited for not administering maintenance therapy, including that the patient had a complete response, the patient desired to travel but would have a lack of access to treatment, the patient was refractory to neoadjuvant chemotherapy, and the patient elected to discontinue maintenance therapy after 3 months. The mean age of this subgroup was 70 years of age; half (50%) had two or more comorbidities, most (79%) had tumor stage T3 or T4 at diagnosis, and most (71%) received fist-line gemcitabine plus carboplatin.

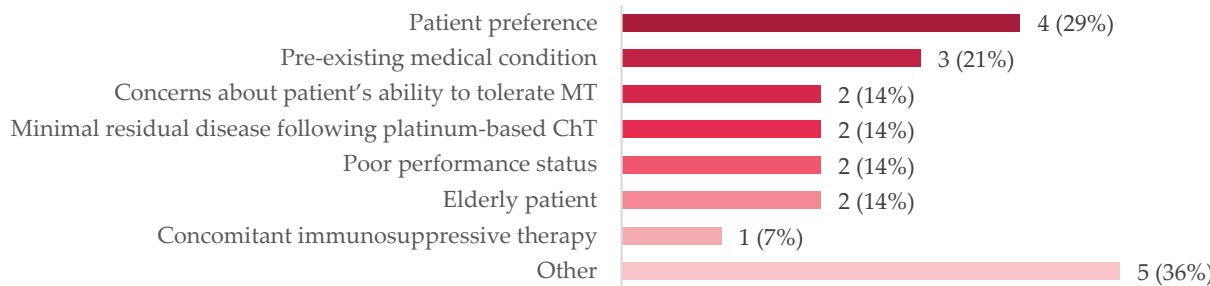

**Figure 1.** Reasons for not administering MT to patients on active surveillance, *n* (%). The following were other reasons that were cited for not administering MT: a patient who chose to discontinue avelumab after 3 months; a patient who wanted to travel after the completion of ChT and the q2wk schedule was not palatable; that there was no reimbursement in 2021; a patient who was refractory to NACT. Abbreviations: ChT = chemotherapy; MT = maintenance therapy; NACT = neoadjuvant chemotherapy; q2wk = every two weeks.

3.3.5. Maintenance Therapy

In total, 85 patients (58%) in the entire cohort were currently receiving maintenance therapy or had prior experience with it. Of these patients, more had received first-line gemcitabine plus cisplatin (58%) than had received first-line gemcitabine plus carboplatin (41%), suggesting that patient fitness (i.e., cisplatin eligibility) is associated with a greater likelihood of receiving maintenance therapy; this may also be an indicator of the differential efficacy profiles of platinum-based regimens. The level of evidence is marginally stronger for cisplatin-based chemotherapy (Level IA recommendation) compared to carboplatin-based chemotherapy (Level IIB recommendation) [12]. Nevertheless, avelumab has demonstrated survival benefits in both of these subgroups [20].

The timeframe between the end of chemotherapy and the start of maintenance therapy was typically 4–6 weeks (52%), although a high proportion started within 6–10 weeks (35%). During the focus groups, it was remarked that 4–6 weeks is a "sweet spot" because it gives just enough time for patients to recover from chemotherapy, but not so long that outcomes may be compromised or that the patient "may get too comfortable off therapy".

In this same subgroup, the maintenance-therapy-emergent adverse events that were difficult or time-consuming were infusion-related reactions and pruritis, as reported in relatively few patients (9%). Most (80%) of these adverse events were managed with supportive treatments (e.g., corticosteroids), and, in some cases, a dose was withheld (40%) or the patient was referred to another specialist for further assessment (20%).

The multidisciplinary team members involved in caring for patients on maintenance therapy included clinic nurses (78%), pharmacists (72%), and infusion nurses (42%) (Table S4 in the Supplementary Materials). Their main responsibilities were to monitor patients for the signs and symptoms of infusion-related reactions, counsel patients after the diagnosis of advanced disease, monitor for treatment-emergent adverse events, and offer ongoing counsel during treatment (Table S4).

Although a large subgroup was on maintenance therapy (46%) at the index date, a small cohort (12%) had received prior maintenance therapy but had subsequently exhibited progressive disease. Of these patients, the median duration of maintenance therapy was 6 months (a range of <1 to 19 months) (Figure 2). In contrast, among all treated patients in the JAVELIN-100 trial, the median duration of trial treatment was 24.9 weeks (range, 2 to 160) in the avelumab group [19]. The median duration of treatment in our study likely reflects a higher-risk population with a more aggressive disease biology who have been treated with maintenance therapy during this initial period of approval. Most (94%) had discontinued maintenance therapy for progressive disease; one (6%) had discontinued therapy due to patient preference.

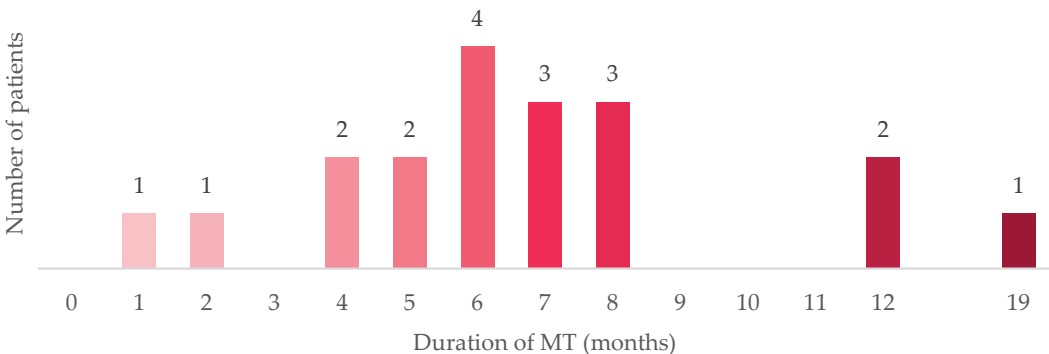

**Figure 2.** Duration of MT before discontinuing therapy. The figure illustrates the duration of MT for the 18 patients in the study who had received MT and had progressed to 2 L therapy or beyond at the index date. Abbreviations: 2 L = second-line therapy; MT = maintenance therapy.

### 3.3.6. Second-Line Therapy or Beyond

In the subgroup of patients on second-line therapy or beyond (*n* = 28, 19%), pembrolizumab or an immunotherapy alternative (avelumab or durvalumab) was the most common treatment option (39%), followed by enfortumab-vedotin (32%), reinduction of chemotherapy (4%), erdafinitib (4%), and a clinical trial (4%) (Table 3). Of those who received second-line immunotherapy, none had prior immunotherapy exposure (i.e., with maintenance therapy). Of the 18 patients whose disease had progressed on MT, none were treated with second-line immunotherapy, but most received enfortumab-vedotin (50%), which is consistent with clinical practice guidelines and quality of evidence [12,13]. When queried about what third-line therapy was being planned for patients still on second-line therapy, most said it was too early to tell or that they had not planned that far ahead (Table S4). Of those who did progress to a third line of therapy (*n* = 12, 8%), the preferred option was administering enfortumab-vedotin (33%), followed by paclitaxel (17%). A subset of patients was treated with metastasis-directed radiation therapy (MDRT) for oligoprogression, but its timing was not captured in this study.

For patients in the other subgroups who had yet to experience progressive disease after first-line therapy, most also indicated that it was too early to tell or that they had not planned that far ahead.

### 3.4. Provider Reflections

Participating oncologists were asked to reflect on their biggest challenges in managing aUC in the context of maintenance therapy (Table 4). They indicated that a lack of institutional resources to monitor and manage patients on maintenance therapy was their biggest challenge (39%). Other challenges included patient reluctance for further treatment (36%) and the management of treatment-emergent adverse events (32%). They also committed to implementing specific actions as a consequence of their participation in the chart review study. Although many (56%) said that the chart review validated their current practice, a significant proportion (54%) said that they would connect with their peers to discuss quality improvement measures resulting from the practice assessment, while some would review the latest guidelines (24%), and others committed to learning more about the management of adverse events associated with maintenance therapy (21%).

**Table 4.** Provider reflections on their practice.

| Reflection (*n* = 17) | *n* | % |
|---|---|---|
| **Biggest challenges with MT** | | |
| Lack of institutional resources (e.g., the ChT clinic already being at capacity) to monitor and manage patients on MT | 57 | 39 |
| Patient reluctance for further treatment (or frequency of treatment) | 52 | 36 |
| Managing treatment-emergent AEs | 46 | 32 |
| Patient or resource factors limiting the ability to start MT within 10 weeks | 8 | 6 |
| Lack of experience with MT | 5 | 3 |
| Lack of access to MT | 0 | 0 |
| **Specific actions to implement in coming months** | | |
| This chart review has validated my current practice | 81 | 56 |
| I will connect with peer(s) to discuss what I have learned in this chart review | 78 | 54 |
| I will review the latest guidelines | 35 | 24 |
| I will learn more about the management of AEs with MT | 30 | 21 |
| I will attend more educational programs to increase confidence in the management of these patients | 26 | 18 |
| I will gain experience with MT | 25 | 17 |

Abbreviations: AEs = adverse events; ChT = chemotherapy; MT = maintenance therapy.

## 4. Discussion

The aim of this real-world retrospective study was to examine treatment patterns among patients with aUC to assess unmet needs in Canadian patients with aUC. To accomplish this, de-identified chart review data of 146 patients with aUC, entered by 17 oncologists across Canada, were collected and analyzed. The baseline demographics and clinical characteristics of patients in this study appeared similar to data from patients enrolled in other real-world studies [27–29]. Clinically relevant data points emanating from this analysis are discussed below, including points about patient characteristics, cisplatin eligibility, maintenance therapy vs. surveillance, and safety considerations.

In terms of patient characteristics, one peculiar finding was the high percentage of patients who presented with de novo metastatic disease (53%), which appeared to be concentrated in Western Canada—participants felt that this was higher than their clinical experience, with more patients typically presenting with the recurrence of early-stage disease. In the focus groups, respondents pointed to the following possibilities: (1) there are differences in referral bases across Canada (which may explain the geographical differences); (2) the entry criteria of the study were skewed for higher-risk, more progressive patients; and (3) the COVID-19 pandemic influenced timely access to care. The impact of the COVID-19 pandemic on the spectrum of cancer care, including delaying diagnoses and treatment, is well documented [30–33]. For example, provincial breast cancer screening programs were suspended for several months during the pandemic [32]. Also, in a review article concerning patients with urological malignancies, it was found that the pandemic dramatically affected patients' access to screening programs and follow-up visits [34]. A survey commissioned by the Canadian Cancer Survivor Network (CCSN) found that 54% had their cancer-care appointments canceled, postponed, or rescheduled because of COVID-19. The same finding was true for approximately 75% of prediagnosis and recently diagnosed patients [35]. In another Canadian study, patients reported feeling dehumanized while receiving their medical care for breast cancer [33]. Taken together, a higher proportion of patients presenting with metastatic disease in our analysis would be consistent with this

interruption in care, the pandemic having coincided with our retrospective timeframe (2021 to 2023).

The chart review also highlighted the variability in assessing eligibility for cisplatin. First, many participants said that they prefer cisplatin over carboplatin because it is perceived as a more efficacious option. Cisplatin-based regimens have historically shown higher response rates than carboplatin-based regimens [17], but a recent update of the JAVELIN-100 trial showed that patients treated with gemcitabine plus cisplatin, followed by maintenance therapy, had similar survival outcomes to those treated with gemcitabine plus carboplatin [20]. Second, participants have a lower creatinine clearance threshold than for the Galsky criteria (creatinine clearance < 60 mL/min for cisplatin ineligibility) [21], with over two-thirds (70%) of participants specifying that they use a threshold of 50 mL/min. The participants indicated that in clinical practice, the Galsky criteria are used to guide treatment selection, but exceptions are routinely considered. This finding is consistent with the Canadian aUC guidelines, which recommend that in select cases, eligibility criteria may be extended to patients with a glomerular filtration rate (GFR) of 45–60 mL/min and/or an Eastern Cooperative Oncology Group (ECOG) performance status of 2 [10]. Split-dose cisplatin is a recommended option for these patients [10,36]. Third, participants noted that they had mixed experiences with regard to cisplatin tolerability at lower creatinine clearance levels. Of those participants (18%) who applied a 60 mL/min cut-off, their preference was driven by patients with marginal creatinine clearance levels who experienced tolerability issues with cisplatin. One of the most important causes of renal dysfunction in patients with aUC is urinary obstruction and hydronephrosis, which is reported in about 25% of patients [37]. This is associated with inferior survival outcomes and a high risk of metastatic potential [38]. The management of hydronephrosis in the context of bladder cancer may include TURBT, percutaneous urinary drainage, and ureteric stenting. The management of urinary obstruction is crucial in those patients who are considered for cisplatin-based chemotherapy [36].

Other participants emphasized the importance of careful patient selection when considering patients with marginal creatinine clearance levels. It was also reasoned that estimated creatinine clearance levels, particularly when marginal, should not be viewed in isolation because the estimating equations correlate poorly with measured renal function [36]. Some participants recommended estimating more than one value for renal function in marginal cases to make a more informed decision. It was also noted that the risk of acute kidney injury may be overstated in marginal cases, given contemporary approaches to managing nephrotoxicity risk.

Participants also discussed their preferences and rationale behind the number of cycles of chemotherapy chosen. Generally, 4–6 cycles are recommended [10], but in practice, most clinicians favor 4 cycles of treatment. The reasons cited for this included sparing the bone marrow for future lines of marrow-depleting agents (such as enfortumab-vedotin), avoiding patient burnout when on chemotherapy prior to maintenance therapy, and there not being enough benefit vs. risk in extending the therapy to 6 cycles. If a patient responds well to 4 cycles of chemotherapy, rechallenging them with chemotherapy in a later line was viewed as a good option. In general, cisplatin is the preferred option if indicated, but it affords less flexibility with regard to extending the treatment from 4 cycles to 6 cycles in cases of high tumor burden; in these instances, carboplatin is often preferred.

An important care gap that we identified from the analysis was the proportion of patients (10%) currently under surveillance or BSC. Most of these patients had either stable disease, a partial response, or a complete response to first-line chemotherapy, and would have been eligible for maintenance therapy on this basis. Although some patients were noted as being ineligible for maintenance therapy (e.g., elderly, poor performance status, or concomitant medications), others opted for BSC based on patient preference. Others also indicated that maintenance therapy was not chosen because there was no residual disease, or because the patient had a complete pathological response, yet maintenance therapy is indicated in all patients whose disease has not progressed following first-line platinum-

based chemotherapy. Patients with a complete response benefited from maintenance therapy in the JAVELIN-100 trial, with a hazard ratio for overall survival of 0.72 [95% CI, 0.48–1.08]. Lack of reimbursement was also cited as a reason for not offering maintenance therapy, although the treatment was accessible for compassionate use after it was approved. Another participant indicated that a patient wanted to travel after completion of the chemotherapy and that the treatment schedule was not amenable. Maintenance therapy is now a standard of care that offers a 9-month survival benefit compared to BSC and all eligible patients should be offered and encouraged to take this treatment [20]. Our study suggests that there may be a lack of awareness about the evidence and recommendations for maintenance therapy.

This study also captured qualitative data about maintenance therapy-related adverse events that were difficult or time-consuming to manage. Participants reported that only a small percentage (7%) of patients on maintenance therapy had challenging adverse events to manage. This is in contrast to the reported adverse events (47.4%) of grade 3 or higher reported in the JAVELIN-100 trial [19]. This discrepancy suggests that these adverse events are manageable. In fact, participants in the focus groups remarked that clinicians have become adept at mitigating and managing adverse events such as infusion-related reactions (reported in 10.2% of patients). In line with prescribing recommendations, most participants premedicate patients with antihistamine and acetaminophen prior to the first 4 avelumab infusions. It was also suggested to reintroduce premedication in patients who take a treatment break for several weeks—for example, to travel. However, participants noted that other common adverse events, like pruritis, can be difficult to manage. Moreover, one-third (32%) of the participants expressed the opinion that managing treatment-related adverse events is one of their biggest challenges in relation to maintenance immunotherapy.

The need for a multidisciplinary team in the management of aUC was not clearly highlighted in our study, despite accumulating evidence that supports the role of con-solidative surgical intervention for low-volume disease, following a good response to chemotherapy [39]. Conversely, MDRT for oligoprogressive disease is emerging as a strat-egy for effective disease control, with data suggesting improved survival outcomes [40].

The reflective component of the chart review questionnaire also supports the notion that there are care gaps in the management of aUC in Canada. For example, two-fifths (39%) of participants said that there is a lack of institutional resources to monitor and manage patients who are on maintenance immunotherapy. In some cases, there were factors limiting the ability to achieve the timely initiation of maintenance immunotherapy within 10 weeks of completing first-line chemotherapy. Some participants even said that they lacked experience with maintenance immunotherapy. In the focus groups, participants also expressed a need for structured guidance on the management of the adverse events associated with maintenance immunotherapy.

There is a scarcity of real-world analyses looking into aUC treatment patterns following the approval of maintenance avelumab treatment. In our study, approximately 50% of the patients received cisplatin-based chemotherapy and the ORR was 70%. In a separate study conducted in France, the United Kingdom, and the United States, with a similar study design, but, prior to the approval of maintenance avelumab, the findings were comparable: 55% of patients received cisplatin-based chemotherapy, the ORR was ~75%, and ~40% of patients received subsequent second-line immunotherapy [29].

In a study conducted in Hungary, 86% of patients who were eligible for systemic treatment received platinum-based chemotherapy, with 78% of these patients receiving cisplatin-based chemotherapy. Overall, few of the treated patients received immunotherapy (6% in the first line, and 23% in the second line). Notably, maintenance avelumab was ap-proved in Europe at the time of the study but was yet to be reimbursed in Hungary [41]. In a study by Swami et al. looking into selected real-world studies, a significant underutilization of first-line treatment was noted, with only 48% of patients receiving treatment [42].

Recently, two phase III trials presented during the European Society for Medical Oncology (ESMO) Congress 2023 showed an OS benefit for patients with aUC, suggesting

a potential shift in the management outlook for aUC in the first-line setting. The EV-302/KEYNOTE-A39 trial demonstrated a doubling of OS rates with the combination of enfortumab-vedotin plus pembrolizumab [43], while the CheckMate 901 trial also showed improved OS results—albeit more modest—with the addition of nivolumab to platinum-based chemotherapy [44]. Nevertheless, continuing to navigate the current standard of care with the use of maintenance avelumab is critical.

With enfortumab-vedotin plus pembrolizumab being anticipated to replace current first-line treatments, patients who are not eligible for this combination may still be candidates for platinum-based chemotherapy and maintenance avelumab. This arrangement is pending further data from CheckMate 901. The addition of immunotherapy to platinum-based chemotherapy has only improved OS in those patients receiving the cisplatin doublet, not the carboplatin doublet. Consequently, carboplatin-based chemotherapy and maintenance avelumab would still have a place in therapy regimens for those who are ineligible for cisplatin treatment.

While platinum-based chemotherapy may be shifting towards a second-line option, the selection of cisplatin or carboplatin informed by Galsky's criteria, other comorbidities, and patient preference will remain vital. However, it is important to note that the effectiveness of maintenance PD-L1 treatment following progression on a PD-1 inhibitor remains uncertain, as the current data do not support this approach [45].

The current first-line treatment recommendation is platinum-based chemotherapy for eligible patients, with both cisplatin-based and carboplatin-based regimens demonstrating improved OS in patients who subsequently receive maintenance avelumab after 4–6 cycles of chemotherapy. Those patients who are ineligible for first-line platinum-based chemotherapy may opt for enfortumab-vedotin plus pembrolizumab, as this combination has shown improved ORR compared to enfortumab-vedotin alone. For individuals anticipating difficulty in tolerating the toxicity of enfortumab-vedotin, single-agent pembrolizumab has demonstrated improved ORR, with a median OS of 11 months.

Approximately 15–20% of patients are expected to undergo a second line of therapy [42], and the choice of such therapies hinges on factors like patient comorbidities, functional status, eligibility, prior immunotherapy usage, the presence of FGFR alterations, and patient preferences.

In anticipation of regulatory approval for enfortumab-vedotin plus pembrolizumab in Canada, our findings emphasize the importance of increased awareness regarding the current first-line standard of care in aUC and strategies to optimize the management of patients on maintenance immunotherapy, potentially leading to improved patient outcomes.

## 5. Limitations

As these findings only represent the early approval period of maintenance immunotherapy, further follow-up is required to confirm if these patterns persist over time. A longer time frame will provide more meaningful real-world insights on treatment patterns, including the utilization of upstream and downstream treatments for patients who may best respond to a maintenance therapy strategy, the sourcing of long-term data to better understand patient outcomes, and an improved treatment duration of maintenance therapy.

Additionally, these data are mostly derived from a small sample of academic/university-affiliated oncology practices, which may limit the generalizability of the outcomes. How the findings from this study would translate to community practice is unknown, but, given that the adoption of new agents in community settings often trails academic practice, we speculate that a greater care gap in aUC management would emerge.

While the chart review questionnaire allowed for a representative portrait of treatment patterns in clinical practice, several limitations exist with these data. Several clinical- and disease-related variables of interest were not fully captured, such as the Gleason grading, ECOG performance status, creatinine clearance, GFR, hearing loss, heart failure, peripheral neuropathy, and treatment history in patients who progressed from early-stage disease. However, we did use proxy variables (e.g., tumor stage, nodal stage, comorbid conditions, and frailty), to partially remediate this issue and have some reflection of cancer severity.

Finally, the study may have been subject to selection bias due to its observational and retrospective design. This bias could arise from factors such as the availability of medical records, the process of choosing patients, and the individual treatment choices of oncologists. To mitigate this bias, we could involve a broader range of oncologists from diverse locations and practice settings or employ statistical techniques to correct for any identified biases. Additionally, a few patients (6%) did not fulfill the patient eligibility criteria, specifically, the fourth criterion about the disease not having progressed while on first-line platinum-based chemotherapy. The purpose of this criterion was to eliminate those candidates who would have been ineligible for maintenance immunotherapy. Despite these limitations, this study is a stepping-stone to understanding practice patterns and prescriber behaviors in the real world.

## 6. Conclusions

Our findings are the first to elucidate aUC treatment patterns in Canada since the approval of maintenance immunotherapy and offer additional insights into the decision-making processes when treating patients with aUC. Our study highlights a care gap as it relates to awareness about the standard first-line care, which comprises first-line platinum-based chemotherapy followed by first-line maintenance therapy in patients who do not progress on chemotherapy. Furthermore, our findings raise interesting hypotheses, setting the stage for larger-scale and longer-term studies to assess the evolution of treatment patterns in the real world. It also highlights the impact of the pandemic on the high de-novo presentation of metastatic disease and the need to further improve cancer patient care in the event of future pandemics.

With our data showing the underutilization of first-line therapy for aUC, future research should focus on a larger dataset and should target the multidisciplinary team involved in the management of aUC. Further steps maybe implemented to reduce the risk of bias and include both prospective and retrospective data collection, also looking into survival parameters and QOL assessments. Steps to improve the current standard of patient care must be refined and this may be accomplished by frequent updating of the national guidelines, taking measures to enhance communication with the multidisciplinary team and raise awareness in the treating physicians about the latest updates by encouraging national conferences and workshops.

**Supplementary Materials:** The following supporting information can be downloaded at: https://www.mdpi.com/article/10.3390/curroncol31020052/s1, Table S1: Medical chart review questionnaire; Table S2: Discussion guide questions for focus groups; Table S3: Additional clinical baseline characteristics; Table S4: Additional patient management patterns.

**Author Contributions:** Conceptualization and methodology, R.R.S., F.A.M., B.J.E., R.M. and M.P.; validation and formal analysis, R.R.S., F.A.M., B.J.E., R.M. and M.P.; writing—original draft preparation, R.R.S., F.A.M. and C.L.P.; writing—review and editing, R.R.S., F.A.M., C.L.P., B.J.E., R.M. and M.P. All authors have read and agreed to the published version of the manuscript.

**Funding:** This research was funded by EMD Serono Canada, a business of Merck KGaA, grant number 202308.4398.QC.

**Institutional Review Board Statement:** Ethical review and approval were waived for this study by REB as no patient data was collected, no dob or name.

**Informed Consent Statement:** Patient consent was waived as the source of the data was de-identified patient records.

**Data Availability Statement:** Data are contained within the article and the Supplementary Materials.

**Acknowledgments:** Medical writing support was provided by Paul Heron of the liV Medical Education Agency, a division of the Clinical Education Alliance.

**Conflicts of Interest:** F.A.M.: No conflict of interest; C.L.P. No conflict of interest; B.J.E.: BMS honoraria, Janssen honoraria, Pfizer honoraria, EMD Serono honoraria and travel expenses, Seagen

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
