# Peer review of "A Real-World Retrospective Analysis of the Management of Advanced Urothelial Carcinoma in Canada"

_curroncol, doi:10.3390/curroncol31020052_

Round 1

Reviewer 1 Report

Comments and Suggestions for Authors

The authors should be congratulated for their work. The manuscript enlightened the clinical practice in Canada regarding urothelial carcinoma in an advanced setting. Specifically, the manuscript aimed to give a comprehensive overview of aUC treatment patterns in Canada following the approval of maintenance immunotherapy,  The role of immunotherapy is still debated in EAU guidelines and it stands as a gap in the current literature. However, the work is worthy of publication after minor corrections. The current paper added a novel perspective to the topic (including Canadian relevant data on the management of aUC).

  • Despite this, the following suggestions could be addressed to improve the scientific soundness of the whole work:
  • - Are data available on the previous radiotherapy settings, delivered to these patients? Authors should consider this aspect (PMID=35625979);
  •  
  •  

- The FGFR rates of Table 1 should be better explained. Specifically, negative and unknown should be split. Unknown should be coupled with Not tested. However, the remaining tables are well written and easily readable.

Author Response

Thank for your comments, replies are colored in red

  • Are data available on the previous radiotherapy settings, delivered to these patients? Authors should consider this aspect (PMID=35625979)

Additional data about RT has been added, highlighted in yellow in the manuscript

  • The FGFR rates of Table 1 should be better explained. Specifically, negative and unknown should be split. Unknown should be coupled with Not tested. However, the remaining tables are well written and easily readable.

Changes has been made, highlighted in yellow in the manuscript

Reviewer 2 Report

Comments and Suggestions for Authors

Your work reflects a real world assesment of differences in management of advanced urotheliala cancers and unmeet needs.

In the last years new paradigms in oncological treatments of urological cancers are changing the management and perspectives.

However this work have some limitations you correctly underlined and the strenght of open internal expert discussion as focus groups highlighted critcal points.

About platinum eligibility it could be useful to add information about a common urothelial cancer complications as renal impairment associated to hydronephrosis and its management. 

With Cancer progression other clinicians are mandatory in multidisciplinary teams ( as urologist/radiologist etc.).

Can you add information about neoadjuvant treatment in patients underwent to radical cystectomy?

Or any data about derivations and AE in this population.

Finally, about COvid Pandemic you cited breast cancer experience but actually you can cite data about uro-oncological patients more properly.

Gavi F, Santoro PE, Amantea C, Russo P, Marino F, Borrelli I, Moscato U, Foschi N. Impact of COVID-19 on Uro-Oncological Patients: A Comprehensive Review of the Literature. Microorganisms. 2023 Jan 10;11(1):176. doi: 10.3390/microorganisms11010176. PMID: 36677468; PMCID: PMC9865028.

Author Response

Thank for your comments, replies are colored in red

About platinum eligibility it could be useful to add information about a common urothelial cancer complications as renal impairment associated to hydronephrosis and its management. 

Added, highlighted in yellow in the manscript

With Cancer progression other clinicians are mandatory in multidisciplinary teams ( as urologist/radiologist etc.).

Added, highlighted in yellow in the manscript

Can you add information about neoadjuvant treatment in patients underwent to radical cystectomy?

unfortunately, we don't have this data avaliable

Or any data about derivations and AE in this population.

Added, highlighted in yellow in the manscript

Finally, about COvid Pandemic you cited breast cancer experience but actually you can cite data about uro-oncological patients more properly.

Added, highlighted in yellow in the manscript

Reviewer 3 Report

Comments and Suggestions for Authors

The authors have attempted to demonstrate real-world treatment patterns by conducting a retrospective medical chart review of oncologists across Canada who treat advanced or metastatic urothelial carcinoma (aUC) to assess unmet needs in Canadian patients with aUC.

The study is methodologically well performed and is well written, however, the study population is relatively small.

The results of this study have clarified aUC treatment patterns in Canada since the approval of maintenance immunotherapy. The study highlights the gap in care as it is referring to awareness of standard first-line care, which includes first-line platinum-based chemotherapy followed by first-line maintenance therapy in patients who do not progress.

However, even this type of observational analysis would require a larger dataset.   

Author Response

Thank you for your comments

Reviewer 4 Report

Comments and Suggestions for Authors

The paper is valuable however needs some improvement before it may be considered for publication in Current Oncology. 

1. Data may be presented on the charts and plots as it makes the paper more citable. 

2. Ensure the uniformity and completeness of the retrospective data. This could involve developing a more standardized protocol for extracting information from medical charts to minimize variability and potential bias in data interpretation. If you have one please present in M&M section.

3. Acknowledge and discuss the potential for selection bias given the retrospective and observational nature of the study. This could include biases related to which patients are chosen or available for review, as well as the oncologists' personal treatment preferences. Implement strategies to mitigate these biases, such as including a broader range of oncologists from different regions or settings, or using statistical methods to adjust for known confounders.

5. Discuiss / Conduct a comparative analysis with other regions or countries, if possible, to contextualize the findings within a global framework. This would help determine if the patterns observed are unique to Canada or reflect a wider trend in aUC management. Also, compare the real-world data with clinical trial results or guidelines to highlight discrepancies or confirmations in practice patterns.

6. Provide more concrete recommendations or implications for clinical practice based on the findings. While the study highlights variability and challenges, actionable steps or guidelines for oncologists could further enhance the paper's utility. Discuss how the findings might influence future research, policy, or guideline development in the management of aUC.

Author Response

Thank you for your comments, replies are colored in red 

  1. Data may be presented on the charts and plots as it makes the paper more citable.

Added in the manuscript

2. Ensure the uniformity and completeness of the retrospective data. This could involve developing a more standardized protocol for extracting information from medical charts to minimize variability and potential bias in data interpretation. If you have one please present in M&M section.

We did standardize the extraction of data from medical charts by developing a standard questionnaire for oncologists to complete for each patient.

3. Acknowledge and discuss the potential for selection bias given the retrospective and observational nature of the study. This could include biases related to which patients are chosen or available for review, as well as the oncologists' personal treatment preferences. Implement strategies to mitigate these biases, such as including a broader range of oncologists from different regions or settings, or using statistical methods to adjust for known confounders.

Added in the manuscript, highlighted in yellow

5. Discuiss / Conduct a comparative analysis with other regions or countries, if possible, to contextualize the findings within a global framework. This would help determine if the patterns observed are unique to Canada or reflect a wider trend in aUC management. Also, compare the real-world data with clinical trial results or guidelines to highlight discrepancies or confirmations in practice patterns.

Added in the manuscript, highlighted in yellow, data are scarce and most of the data are before the era of manitenance IO

6. Provide more concrete recommendations or implications for clinical practice based on the findings. While the study highlights variability and challenges, actionable steps or guidelines for oncologists could further enhance the paper's utility. Discuss how the findings might influence future research, policy, or guideline development in the management of aUC.

Added in the manuscript, highlighted in yellow

Round 2

Reviewer 3 Report

Comments and Suggestions for Authors

The authors should be commended on this study, I have no further questions.